# Prediction model for recommending coronary artery calcium score screening (CAC-prob) in cardiology outpatient units: A development study

Pakpoom Wongyikul[1], Apichat Tantraworasin[2], Pannipa Suwannasom[3], Tanop Srisuwan[4], Yutthaphan Wannasopha[4], Phichayut Phinyo[1,5,6]*

1 Center for Clinical Epidemiology and Clinical Statistics, Faculty of Medicine, Chiang Mai University, Chiang Mai, Thailand, 2 General Thoracic Unit, Department of Surgery, Faculty of Medicine, Chiang Mai University Hospital, Chiang Mai, Thailand, 3 Division of Cardiology, Department of Internal Medicine, Faculty of Medicine, Chiang Mai University, Chiang Mai, Thailand, 4 Department of Radiology, Faculty of Medicine, Chiang Mai University, Chiang Mai, Thailand, 5 Department of Family Medicine, Faculty of Medicine, Chiang Mai University, Chiang Mai, Thailand, 6 Center of Multidisciplinary Technology for Advanced Medicine, Faculty of Medicine, Chiang Mai University, Chiang Mai, Thailand

* phichayutphinyo@gmail.com

## Abstract

Despite the well-established significance of the CAC score as a cardiovascular risk marker, the timing of using CAC score in routine clinical practice remains unclear. We aim to develop a prediction model for patients visiting outpatient cardiology units, which can recommend whether CAC score screening is necessary. A prediction model using retrospective cross-sectional design was conducted. Patients who underwent CAC score screening were included. Eight candidate predictors were preselected, including age, gender, DM or primary hypertension, angina chest pain, LDL-C ($\geq$130 mg/dl), presence of low HDL-C, triglyceride ($\geq$150 mg/dl), and eGFR. The outcome of interest was the level of CAC score (CAC score 0, CAC score 1–99, CAC score $\geq$100). The model was developed using ordinal logistic regression, and model performance was evaluated in terms of discriminative ability and calibration. A total of 360 patients were recruited for analysis, comprising 136 with CAC score 0, 133 with CAC score 1–99, and 111 with CAC score $\geq$100. The final predictors identified were age, male gender, presence of hypertension or DM, and low HDL-C. The model demonstrated excellent discriminative ability (Ordinal C-statistics of 0.81) with visually good agreement on calibration plots. The implementation of this model (CAC-prob) has the potential to enhance precision in recommending CAC screening. However, external validation is necessary to assess its robustness in new patient cohorts.

## Introduction

Numerous conventional risk prediction tools have been developed to assess the baseline risk of atherosclerotic cardiovascular diseases (ASCVD) and guide the initiation of statin therapy

**Data Availability Statement:** All relevant data are within the manuscript and its Supporting Information files.

**Funding:** The author(s) received no specific funding for this work.

**Competing interests:** The authors have declared that no competing interests exist.

[1–3]. The American College of Cardiology/American Heart Association (ACC/AHA) practice guidelines currently endorse the use of the pooled cohort equation (PCE) [4], whereas the Canadian Cardiovascular Society (CCS) guideline recommends using the Framingham risk score (FRS) [5]. Other prediction tools, such as Systematic Coronary Risk Evaluation (SCORE), QRISK3, and Thai CV risk, have been recommended by national guidelines to specifically estimate ASCVD risk in their respective populations [6–8]. Nevertheless, these ASCVD risk prediction tools have exhibited suboptimal performance in real-world observational studies [9, 10]. Since these tools attempt to forecast the natural progression of the disease, and a significant proportion of patients in these observational studies receive therapies during follow-up, the estimated risk may not match and properly guide statin therapy for those statin-naïve patients [11].

Extensive research over the years strongly supports the potential of Coronary Artery Calcium (CAC) score as a valuable tool for cardiovascular risk stratification [12]. Not only does the CAC score outperform conventional risk scores independently, but it also enhances their discriminative capacity in predicting ASCVD risk [12, 13]. Furthermore, the visual impact of individual plaque burden indicated by the CAC score has the potential to improve patient compliance with prescribed therapies [14]. In a recent randomized controlled trial comparing CAC score-based statin guidance to the Pooled Cohort Equation (PCE) for primary ASCVD prevention, CAC score-based guidance demonstrated superior LDL-C reduction and potentially greater efficiency in statin dispensing [14].

Despite the well-established effectiveness of the CAC score as a valuable cardiovascular marker, its widespread implementation faces significant challenges in various settings, including Thailand. Most concerns hindering the implementation of CAC screening still revolve around the expected increase in healthcare costs and unnecessary radiation exposure from computed tomography (CT) scans. Another substantial barrier is the lack of evidence demonstrating that statin use based on the CAC score leads to improvements in hard clinical outcomes [15]. According to the Thailand Clinical Practice Guideline on Pharmacologic Therapy of Dyslipidemia for ASCVD Prevention in 2016, the use of CAC score for risk stratification was not yet recommended [16]. Consequently, the timing of using CAC score as an adjunctive risk classifier in routine clinical practice remains unclear and is left to physician discretion. For these reasons, the decision to recommend CAC score screening must be accurate. Our objective was to develop a prediction model that can recommend whether patients should undergo CAC score screening.

## Methods

### Study design and patients

We developed a prediction model using retrospective cross-sectional data. The study included patients who underwent CAC score screening due to suspicion of having CAC score >0, as determined by their attending physicians at Maharaj Nakorn Chiang Mai Hospital between January 2012 and December 2018. Patients without an official CAC score report and those with a history of major adverse cardiovascular events (MACE) were excluded. MACE was defined as a composite of cardiovascular death, myocardial infarction (MI), heart failure-related hospitalization, coronary artery revascularization procedures, ischemic stroke, or transient ischemic attack [17]. The study received ethical approval from the Institutional Review Board and Ethics Committee of the Faculty of Medicine, Chiang Mai University (HOS 2566–0094). Informed consent is not required.

## Data collection

Demographic data (age and gender) and clinical characteristics (smoking status, co-morbidity, symptoms of chest pain) were collected on the date of screening. Laboratory parameters (lipid profiles, serum creatinine, estimated glomerular filtration rate [eGFR]) were obtained on the closest date to the screening, with a maximum interval of 30 days. The presence of previous MACE was reviewed. All data were retrieved from standard electronic medical records. Data were collected from June 2023 to August 2023. Patient identity was accessed only during the data collection and was not collected.

## CAC score

The CAC score was determined through the use of a semi-automatic non-contrast prospective electrocardiogram (ECG) gating scan of the heart, performed with a state-of-the-art 192-slice Dual Source CT scanner (Somatom Force, Siemens). This involved conducting a systolic ECG-Triggered Sequential Shuttle mode scan with a delay of 280–320 milliseconds. The high-pitch scan technique was employed when a patient's heart rate was below 75 beats per minute. The acceleration of the table and initiation of data collection were determined based on the analysis of multiple heartbeats before data collection. The most cranial section was set at 65% of the R-R interval in all patients. A 3 mm slice thickness was used, along with a tube voltage of 120 kV. Certified radiologists applied the Agatston scoring method to measure the extent of CAC score [18].

## Candidate predictors

Eight candidate predictors were preselected based on their previously proposed association with CAC score and/or with major adverse cardiovascular events (MACE) [19–24] and their availability in the outpatient department (OPD). These predictors include age, gender, diabetes mellitus (DM) or primary hypertension (either presence), low-density lipoprotein cholesterol (LDL-C) ($\geq$130 mg/dl), high-density lipoprotein cholesterol (HDL-C) (<40 mg/dl in males, <50 mg/dl in females), triglyceride ($\geq$150 mg/dl), estimated glomerular filtration rate (eGFR) (each 10 ml/min/1.73 m$^2$). Based on our clinical context, encountering patients with no known coronary artery disease (CAD) presenting with symptomatic chest pain, suggestive of stable ischemic heart disease (SIHD) is common [25]. To assess the utility of this factor in predicting CAC score, symptomatic chest pain (presence) was included as a candidate predictor.

## Sample size estimation

To estimate the required sample size for developing a multivariable prediction model with polytomous outcomes, we followed the criteria outlined by Riley et al. [26]. We set the values for Nagelkerke $R^2$, the shrinkage factor, and the maximum allowable absolute difference between the apparent and adjusted Nagelkerke $R^2$ at 0.15, 0.9, and 0.05 respectively. Accordingly, the minimum sample size required was determined to be 328 patients, with 132 patients with CAC score = 0, 99 patients with CAC score 1–99 and 99 patients with CAC score $\geq$ 100.

## Statistical analysis

All statistical analyses were conducted using Stata 17 (StataCorp, Lakeway, Texas, USA). Categorical variables were described using frequencies and percentages. Numerical data were assessed for distribution through histograms and summarized using means and standard deviations (SD) or medians and interquartile ranges (IQR), depending on their distributions. The Wilcoxon non-parametric test for trend [27] was employed to test the difference among all

predictors across the three CAC score categories (CAC score 0, CAC score 1–99, CAC score ≥100) [4, 28]. For each predictor, we assessed its association with CAC score using univariable ordinal logistic regression. If the log odds ratio (OR) remained consistent across the categories, indicating satisfaction of the proportional odds assumption, the proportional odds model was used. In cases where the proportional odds assumption was not met, the partial proportional odds model was employed. Statistical significance was defined at a p-value less than 0.05.

### Model development and missing data handling

The model was developed using the partial proportional odds model with a stepwise backward elimination approach. The outcomes used in the model were CAC score 0, CAC score 1–99, CAC score ≥100. Initially, all pre-selected predictors were included. Subsequently, variables that were not significant at the level of 0.05 were excluded. The probabilities of being each category were estimated. To account for missing data, we employed multiple imputation with chained equation (MICE) [29, 30]. The determination of the number of imputed datasets followed a two-stage approach, ensuring the replicability of standard error [31]. For generating the imputed datasets of continuous variables, predictive mean matching (PMM) with K-nearest neighbour (where K = 10) was used. Binary logistic regression was employed to impute binary variables. CAC score was used to aid in estimating the uncertainty of the imputed datasets.

### Model performance and internal validation

The model performance was measured in terms of discriminative ability and calibration. The model's discriminative ability was evaluated with the average dichotomous C-index, generalized C-index, and ordinal C-index (ORC) [32]. ORC, accounting for the ordinal nature of the outcome, served as the primary measure to describe the discriminative ability of the model. An area under the receiver operating characteristic curve (AuROC) of 0.70–0.80, 0.80–0.90, and above 0.90 was considered acceptable, excellent, and outstanding, respectively [33]. For model calibration, we examined the agreement between predicted events and observed events of lower pairs (CAC score 0 vs CAC score >0) and higher pairs (CAC score <100 vs CAC score ≥100) using calibration plots.

Internal validation was assessed via bootstrap re-sampling with 1000 replicates. C-statistics from lower pairs and higher pairs were then used to assess model optimism. The heuristic shrinkage value was calculated using Van Houwelingen's method [34].

### Cut point selection and model presentation

The diagnostic indices, including sensitivity, specificity, positive predictive value (PPV), negative predictive value (NPV), positive likelihood ratio (LR+), and negative likelihood ratio (LR-), were calculated for two cut points: lower pairs cut point and higher pairs cut point. Our aim was to determine the cut point values that can effectively recommend patients to either undergo CAC screening or not, ensuring a balance between the potential downsides of overusing CAC score screening and its benefits.

For practicality, we transformed the CAC score prediction model into a user-friendly web application called 'CAC-prob.' Once clinical and laboratory parameters are entered, the application calculates the probability of CAC score >0 and CAC score ≥100. Patients were then classified into three distinct ordered categories as follows: (1) Low risk of having a CAC score >0, (2) Moderate risk of having a CAC score >0 but unlikely to be ≥100, and (3) High risk of having a CAC score >0 and likely to be ≥100. Patients in the first group would not be recommended for CAC score screening and would follow routine standard care. Those in the second

group would receive a recommendation for CAC score screening, with physician discretion or an increase in statin potency if they do not meet the reduction target. Patients in the third group would be strongly advised to undergo CAC score screening, along with an increase in statin potency if they do not meet the reduction target. To assist physicians in guiding patient management, the application provides the three recommendations mentioned above for each individual patient based on their individual risk profile.

## Results

### Patient characteristics

From January 2012 to December 2018, a total of 425 patients underwent CAC score screening. Of this total, 65 patients were excluded—34 due to prior occurrences of major adverse cardiovascular events (MACE) and 31 due to the absence of an official CAC score report (Fig 1). The remaining 360 patients were included for analysis, comprising 136 with a CAC score of 0, 133 with CAC score ranging from 1 to 99, and 111 with CAC score ≥100 (Fig 1). The mean age of the study patients was 61.4±10.6 years, with nearly half being male (46.4%). The prevalence of diabetes mellitus (DM), hypertension, dyslipidemia, and chronic kidney disease was 13.9%, 49.7%, 49.4%, and 8.9%, respectively (Table 1). Approximately 40% of all patients were taking antihypertensive (40.8%) and lipid-lowering medications (36.7%). All patients on lipid-lowering drugs were receiving statin therapy.

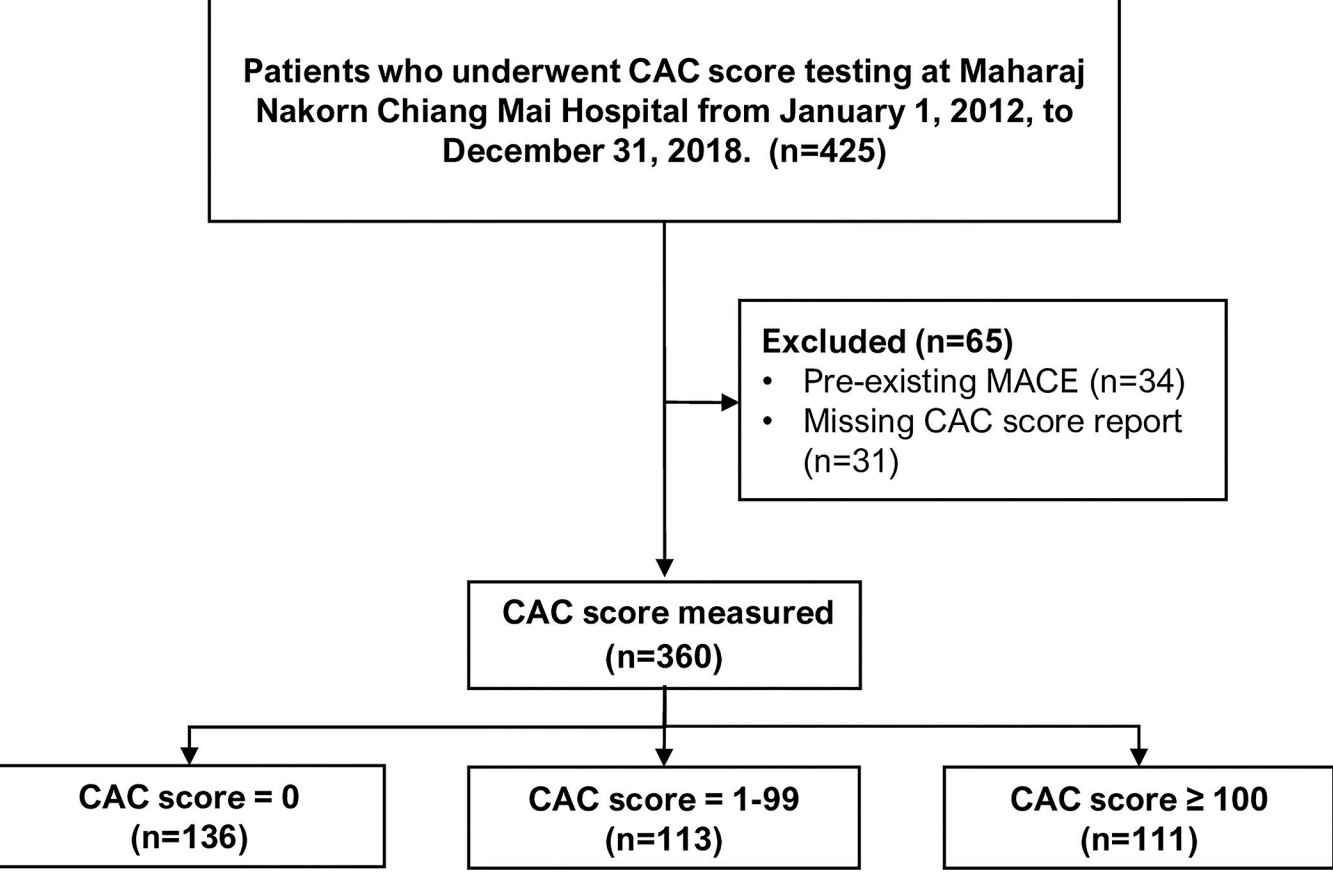

**Fig 1. Study flow diagram.** Abbreviation: CAC, coronary artery calcium; MACE, major adverse cardiovascular event.

**Table 1. Comparison of baseline characteristic between absence and presence of CAC score.**

| Prognostic factors | Missing n (%) | Total (n = 360) n (%) | CAC = 0 (n = 136) n (%) | CAC = 1–99 (n = 113) n (%) | CAC ≥100 (n = 111) n (%) | P value |
|---|---|---|---|---|---|---|
| Age (years) Mean, SD | 9 (2.5) | 61.4 ±10.6 | 56.6 ±9.6 | 62.9 ±9.6 | 65.9±10.2 | <0.001 |
| Male | 6 (1.7) | 167 (46.4) | 44 (32.4) | 62 (54.9) | 61 (55.0) | <0.001 |
| Smoking status | 105 (29.2) | | | | | 0.145 |
| never smoking | | 233 (64.7) | 90 (66.2) | 73 (64.6) | 70 (63.1) | |
| former smoking | | 14 (3.9) | 4 (2.9) | 4 (3.5) | 6 (5.4) | |
| current smoking | | 8 (2.2) | 2 (1.5) | 2 (1.8) | 4 (3.6) | |
| DM | 68 (18.9) | 50 (13.9) | 17 (12.5) | 10 (8.9) | 23 (20.7) | 0.137 |
| Hypertension | 69 (19.2) | 179 (49.7) | 54 (39.7) | 52 (62.0) | 73 (65.8) | <0.001 |
| Antihypertensive drug | 77 (21.4) | 147 (40.8) | 47 (34.6) | 42 (37.2) | 58 (52.3) | 0.026 |
| HDL-C(mg/dl) mean, SD | 139 (38.6) | 53.8±14.8 | 56.7±16.8 | 51.7±13.2 | 53.1±13.9 | 0.231 |
| LDL-C(mg/dl) mean, SD | 130 (36.1) | 117.1 ±38.1 | 120.0 ±38.8 | 114.5 ±33.4 | 116.5±41.5 | 0.452 |
| Triglyceride median, IQR | 141 (39.2) | 109 (81, 147) | 105 (71.5, 146.5) | 113.5 (88, 148) | 109 (82, 143) | <0.378 |
| Dyslipidemia | 69 (19.2) | 178 (49.4) | 51 (37.5) | 56 (49.6) | 71 (64.0) | <0.001 |
| Lipid lowering drug | 75 (20.8) | 132 (36.7) | 38 (27.9) | 39 (34.5) | 55(49.6) | 0.002 |
| Creatinine (mg/l) mean, SD | 136 (37.8) | 0.95 ±0.54 | 0.83± 0.23 | 0.95 ±0.22 | 1.07±0.83 | <0.001 |
| eGFR (ml/min/1.73 m²) | 138 (38.3) | 80.1 ±19.3 | 87.6 ±18.0 | 77.5 ±16.6 | 75.6±20.8 | <0.001 |
| Chronic kidney disease | 70 (19.4) | 32 (8.9) | 6 (4.4) | 9 (8.0) | 17(15.3) | 0.005 |
| Symptomatic | 59 (16.4) | 140 (38.9) | 53 (39.0) | 45 (39.8) | 42 (37.8) | 0.348 |
| CAC for risk screening | 12 (3.3) | 302 (83.9) | 129 (94.9) | 92 (81.4) | 81 (73.0) | <0.001 |

**Abbreviation**: DM, diabetes mellitus; eGFR: estimated glomerular filtration rate; HDL, high-density lipoprotein; IQR, interquartile range; LDL, low-density lipoprotein; SD, standard deviation

**Table 2. The association between prognosis factors and CAC >0 and CAC ≥100 using univariable partial proportional odds modelling.**

| Prognostic factors | CAC > 0 | | | CAC ≥ 100 | | |
|---|---|---|---|---|---|---|
| | OR (95% CI) | P value | ROC (95% CI) | OR (95% CI) | P value | ROC (95% CI) |
| Age (years) | 1.07 (1.05–1.10) | <0.001 | 0.71 (0.65–0.76) | 1.07 (1.05–1.10) | <0.001 | 0.69 (0.64–0.75) |
| Male* | 2.19 (1.33–2.96) | 0.002 | 0.61 (0.55–0.66) | 1.91 (1.20–3.03) | 0.006 | 0.55 (0.50–0.60) |
| Hypertension or DM | 1.98 (1.27–3.07) | 0.003 | 0.58 (0.53–0.64) | 1.98 (1.27–3.07) | 0.003 | 0.59 (0.54–0.64) |
| low HDL-C | 1.80 (1.08–3.01) | 0.025 | 0.55 (0.51–0.58) | 1.80 (1.08–3.01) | 0.025 | 0.54 (0.50–0.59) |
| LDL-C(mg/dl) ≥130* | 0.90 (0.47–1.43) | 0.468 | 0.53 (0.47–0.60) | 0.85 (0.50–1.43) | 0.540 | 0.49 (0.43–0.55) |
| Triglyceride ≥150* | 0.92 (0.57–1.81) | 0.658 | 0.50 (0.45–0.56) | 0.95 (0.59–1.53) | 0.825 | 0.52 (0.46–0.57) |
| symptomatic | 0.87 (0.57–1.34) | 0.535 | 0.50 (0.44–0.56) | 0.87 (0.57–1.34) | 0.535 | 0.49 (0.44–55) |
| eGFR (10ml/min/1.73 m²)* | 0.70 (0.59–0.83) | <0.001 | 0.69 (0.62–0.77) | 0.78 (0.67–0.90) | 0.001 | 0.61 (0.53–0.68) |

*: **Variable that did not meet a proportional odds assumption. Abbreviation**: DM, diabetes mellitus; eGFR: estimated glomerular filtration rate; HDL-C, high-density lipoprotein cholesterol; LDL-C, low-density lipoprotein cholesterol; OR, unadjusted odds ratio; ROC, pairwise receiving operative characteristic

**Table 3. Multivariable adjusted odds ratio, beta coefficient for the full model and the reduced model using partial proportional odds modelling.**

| Prognostic factors | Full model | | | Reduced model | | |
|---|---|---|---|---|---|---|
| | aOR (95% CI) | P value | beta | aOR (95% CI) | P value | beta |
| Age (years) | 1.08 (1.05–1.11) | <0.001 | 0.07 | 1.08 (1.06–1.11) | <0.001 | 0.80 |
| Male | 2.73 (1.76–4.24) | <0.001 | 1.00 | 2.85 (1.86–4.35) | <0.001 | 1.05 |
| Hypertension or DM | 1.81 (1.08–3.03) | 0.025 | 0.59 | 1.78 (1.09–2.92) | 0.021 | 0.58 |
| low HDL-C(mg/dl) | 2.27 (1.28–4.03) | 0.006 | 0.82 | 2.31 (1.32–4.05) | 0.003 | 0.84 |
| LDL-C (mg/dl) ≥130* | 1.11§ (0.63–1.97) | 0.718§ | 0.10§ | Not included | | |
| Triglyceride (mg/dl) ≥150 | 1.28 (0.72–2.28) | 0.395 | 0.25 | Not included | | |
| symptomatic | 0.84 (0.52–1.36) | 0.477 | -0.17 | | | |
| eGFR (10ml/min/1.73 m²)* | 0.89§ (0.73–1.07) | 0.249§ | -0.12§ | Not included | | |
| Constant at cut > 0 | | | -3.94 | | | -5.32 |
| LDL-C (mg/dl) ≥130* | 1.18 (0.70–1.98) | 0.539¶ | 0.17¶ | | | |
| eGFR (10ml/min/1.73 m²)* | 0.94 (0.76–1.14) | 0.505¶ | -0.07¶ | | | |
| Constant at cut ≥100 | | | -5.99 | | | -6.92 |

*: Variable that did not meet a proportional odds assumption

§: The value was derived from the outcome CAC >0

¶ **The value was derived from the outcome CAC >100 Abbreviation**: aOR, adjusted odds ratio; DM, diabetes mellitus; eGFR: estimated glomerular filtration rate; HDL-C, high-density lipoprotein cholesterol; LDL-C, low-density lipoprotein cholesterol

Table 1 compares the baseline characteristics across CAC score categories. Age, sex, hypertension prevalence, dyslipidemia prevalence, creatinine, and estimated glomerular filtration rate (eGFR) demonstrated a statistically significant increasing trend across the CAC score categories. Similar proportions of patients who reported angina chest pain were observed in all groups. Further details on prognostic factors are presented in Table 1.

## Candidate predictors

Age, presence of hypertension or DM, and low HDL-C were significantly associated with both CAC score >0 and CAC score ≥100 in a proportionate manner (Table 2). Table 2 presents the discriminative ability and univariable association of each candidate predictor with CAC score categories at two cut points (CAC score >0 and CAC score ≥100). Four predictors, including male gender, LDL-C ≥130, Triglyceride ≥150, and eGFR, did not meet the proportional odds assumption. Only male gender and eGFR were significantly associated with CAC score >0 and CAC score ≥100, with the OR between CAC score >0 and CAC score ≥100 showing slight differences (Table 2).

In terms of discriminative ability, three of the significant predictors (age, male gender, eGFR) exhibited pairwise ROC values above 0.6 in both CAC score categories. Overall, the pairwise ROC was slightly higher for CAC score >0 compared to CAC score ≥100 (Table 2). In the full model, only age, male gender, presence of hypertension or DM, and low HDL-C

remained statistically significant. LDL-C ≥130 and eGFR did not meet the proportional odds assumption (Table 3).

## Model performance

In the reduced model, four final predictors were identified: age (OR 1.08, 95% confidence interval [CI]: 1.05–1.11, P-value: <0.001), male gender (OR 2.85, 95% CI: 1.85–4.35, P-value: <0.001), presence of hypertension or DM (OR 1.78, 95% CI: 1.09–2.92, P-value: 0.021), and low HDL-C (OR 2.31, 95% CI: 1.32–4.05, P-value: 0.003) (Table 3). All of the final predictors met the proportional odds assumption. In terms of discriminative ability, the generalized C-index, average C-index, and ORC were 0.73, 0.76, and 0.81, respectively (S1 Table in S1 File). The average ROC of lower pairs and higher pairs were 0.79 and 0.73 (S1 Table in S1 File). The generalized ROC of all pairwise outcomes (CAC score 0 vs CAC score 1–99, CAC score 0 vs CAC score ≥100, and CAC score 1–99 vs CAC score ≥100) are reported in S1 Table in S1 File.

Both calibration plots of the reduced model demonstrated visually good agreement between observed and predicted probability (Fig 2). Internal validation for lower pairs and higher pairs revealed apparent performances of 0.78 and 0.72, bootstrap performances of 0.76 and 0.71, and optimism values of 0.08 and 0.01, respectively. The heuristic shrinkage value was 0.95.

# Calibration plot

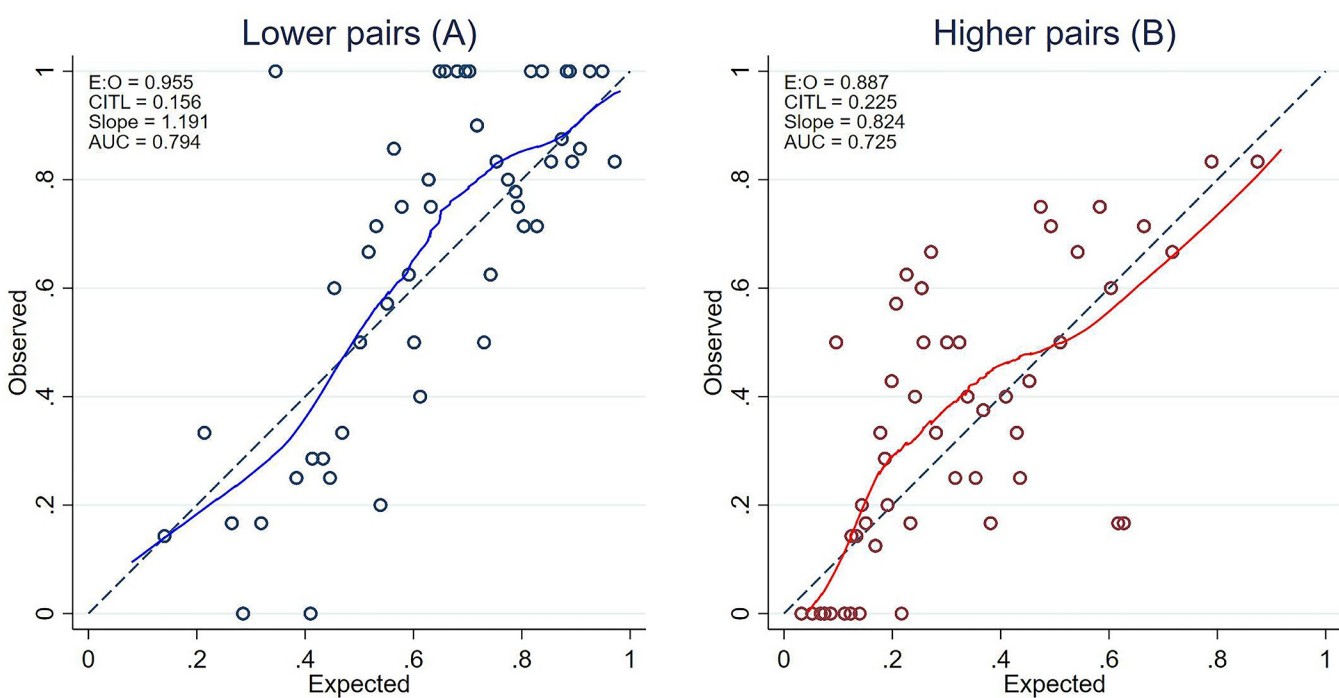

**Fig 2. Calibration plot.** A and B illustrates the calibration plot for the agreement between predicted and observed probability of lower pairs (CAC >0 vs CAC <0) and higher pairs (CAC ≥100 vs CAC ≤100), respectively. Dash line illustrates the perfect prediction line. Solid blue and red lines illustrate calibration curve of lower pairs and higher pairs, respectively. hollow circles represent observed date. Abbreviation: CAC, coronary artery calcium.

**Table 4. Diagnostic index in each cut point.**

| Cut point | Sensitivity (95%CI) | Specificity (95%CI) | PPV (95%CI) | NPV (95%CI) |
|---|---|---|---|---|
| Cut point for CAC screening is recommended (lower pairs cut point) | | | | |
| ≥0.45 | 91.3 (86.6–94.8) | 49.1 (39.3–58.9) | 77.5 (71.7–82.5) | 74.6 (62.9–84.2) |
| ≥0.50 | 88.4 (83.2–92.4) | 57.4 (47.5–66.9) | 79.9 (74.1–84.9) | 72.1 (61.4–81.2) |
| ≥0.55 | 81.2 (75.2–86.2) | 67.6 (57.9–76.3) | 82.8 (76.8–87.7) | 65.2 (55.6–73.9) |
| Cut point for CAC screening is strongly recommended (higher pairs cut point) | | | | |
| ≥0.20 | 86.7 (79.1–92.4) | 48.0 (41.0–55.1) | 48.3 (41.2–55.4) | 86.6 (78.9–92.3) |
| ≥0.25 | 75.2 (66.2–82.9) | 58.4 (51.3–65.3) | 50.3 (42.5–58.1) | 80.8 (73.5–86.9) |
| ≥0.30 | 61.9 (52.3–70.9) | 64.4 (57.3–71.0) | 49.3 (40.8–57.8) | 75.1 (68.0–81.4) |

Specific cut point value was based on the estimated probability of CAC >0. **Abbreviation:** CI, confidence interval; PPV, positive predictive value; NPV, negative predictive value

## Cut point thresholds

To determine the low-risk group, three potential lower pairs cut points were identified, including the probability of CAC score >0 at 0.45, 0.50, and 0.55, demonstrating sensitivities above 80%, with specificities of 49.1% (95% CI 39.3%-58.9%), 57.4% (95% CI 47.5%-66.9%), and 67.6% (95% CI 57.9%-76.3%) (Table 4). To identify patients with a high risk of CAC score >0 from the moderate-risk group, higher pairs cut points at 0.20, 0.25, and 0.30 were considered. These thresholds displayed sensitivities of 86.7% (95% CI 79.1%-92.4%), 75.2% (95% CI 66.2%-82.9%), and 61.9% (95% CI 52.3%-70.9%), accompanied by specificities of 48.0% (95% CI 41.0%-55.1%), 58.4% (95% CI 51.3%-65.3%), and 64.4% (95% CI 57.3%-71.0%), respectively (Table 4).

Detailed information on other diagnostic indices and the proportion of classified patients for specific cut points can be found in S2-S5 Tables in S1 File. The CAC-prob calculator, along with additional information on how the probabilities were calculated, is available in S2 File.

## Discussion

In this study, we developed a prediction model for recommending CAC Score Screening (CAC-prob). The model utilized four routine clinical predictors: age, male gender, the presence of hypertension or DM, and low HDL-C (Fig 3). Notably, the model demonstrated excellent discriminative ability and good calibration. CAC-prob used individual risk factors to provide decisive guidance on the necessity of undergoing CAC score screening for refining risk.

In 2020, a machine learning-based logistic regression using clinical predictors was employed to predict CAC score = 0 and ≥400 [35]. This study highlighted the significant contributions of age and gender to the predictive ability of the model. Although the baseline characteristics of the previous studies were similar to the current analysis, they included patients who may experience MACE, which was discordant with the scope of our study. Additionally, the implementation of those models could be technically difficult due to the nature of machine learning and the complexity of predictors, which was not aligned with our focus on outpatient settings.

In our study, we did not observe an association between LDL-C levels ≥130 mg/dL or triglyceride levels ≥150 mg/dL and CAC score >0 or CAC score ≥100. Surprisingly, our results indicated a modest protective effect. This phenomenon might be attributed to the nature of our cross-sectional design and the inclusion of a mixed population, which included both

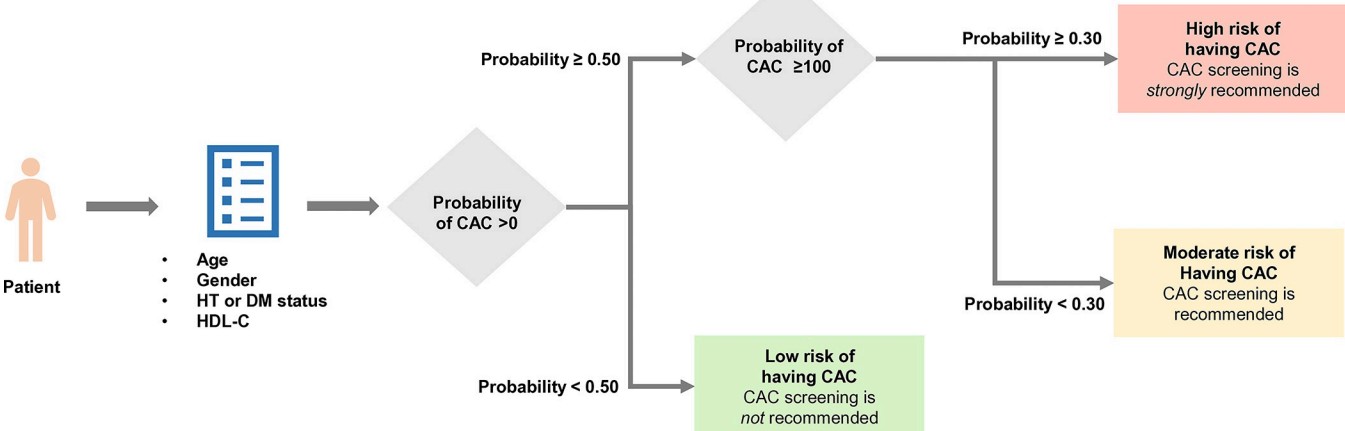

**Fig 3. The recommendation algorithm to implement CAC-prob in practice.** After entering age, gender, HT or DM status, and HDL-C information, patients are initially assessed for their probability of having CAC >0. Patients with a probability <0.50 are categorized as low risk for having CAC. Therefore, CAC screening is not recommended, and they follow routine standard care. For patients with a probability ≥0.50, their probability of having CAC >100 is evaluated. Patients with a probability <0.30 are categorized as moderate risk of having CAC and recommended to undergo CAC screening with physician discretion or to increase statin potency if the patient does not meet the reduction target. For patients with a probability ≥0.30, patients are categorized as high risk of having CAC. CAC screening is strongly recommended, along with an increase in statin potency if the patient does not meet the reduction target. Abbreviation: HT, hypertension; DM, diabetes mellitus; HDL-C, high-density lipoprotein cholesterol; CAC, coronary artery calcium.

patients who were or were not statin-naïve. Notably, patients with higher CAC score tended to receive intensive statin therapy [36]. This finding was also observed in other previous studies [20, 37]. On the other hand, low HDL-C indicated an independent association with CAC score >0 and CAC score ≥100. These findings were consistent with a recent MESA cohort study involving non-naïve statin patients [21]. The study showed an inverse association between HDL-C and CAC score >0 and CAC score ≥100 in cross-sectional analysis, and cohort perspective as well [21]. The MESA study also revealed that lipid-lowering therapy did not modify the association between HDL-C and CAC score progression [21]. The effect of statins on HDL-C was relatively small compared with LDL-C [38]. This property of HDL-C underscores its strength as a stable predictor, making it well-suited for OPD.

For our development set, patients with DM, accounting for 15% of the total population, were included in the analysis. Due to the inherently high atherosclerotic cardiovascular disease (ASCVD) risk among DM patients, most of them would probably be treated with statin therapy irrespective of their CAC score. However, the added value of CAC score screening, as recommended by CAC-prob, lies in its ability to enhance discriminative ability and reclassify risk upward regardless of DM status [39, 40]. In cases where DM patients have CAC score ≥100, there is an opportunity for risk reclassification, potentially leading to a more aggressive statin therapy approach and subsequent risk reduction [39, 41, 42].

It is crucial to note that CAC serves as an adjunctive ASCVD risk assessment rather than as a stand-alone test. Our goal was to encourage the utilization of the CAC score while ensuring a balance between the costs and complications associated with unnecessary statin dispensing and the overuse of CAC score assessments. CAC-prob aims to support those who may benefit from a CAC score to guide statin therapy. This includes not only statin-naïve patients with intermediate risk [4] but also high-risk patients or those with risk enhancers such as diabetes or hypertension, who are reluctant to accept treatment or whose clinicians are concerned about intensifying statin therapy [43]. We proposed a threshold of 0.50 for the lower cut point to emphasize high sensitivity and a threshold at 0.30 for the higher pairs cut point aiming to

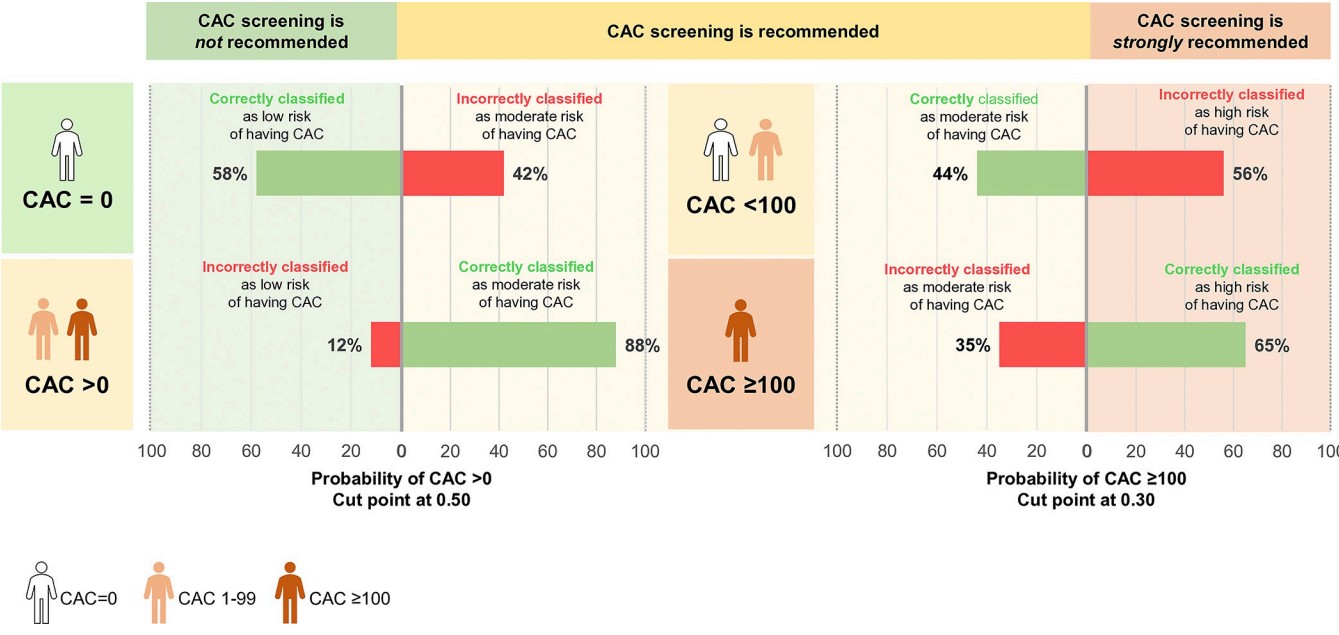

**Fig 4. The classification accuracy of CAC-prob for the lower and higher cut points.** When assessing the probability of having CAC >0, 12% of patients with CAC >0 are incorrectly classified as low risk for having CAC, while 88% are correctly classified as moderate risk for having CAC. On the other hand, 42% of patients with CAC of 0 are incorrectly classified as moderate risk for having CAC. When assessing the probability of having CAC >100, 65% of patients with CAC ≥100, who were correctly classified from the lower pairs cut point, are correctly classified as high risk for having CAC, while 35% are incorrectly classified and remain in the moderate risk group. The correct classification proportion of patients with CAC <100, including patients with CAC 1–99 and CAC of 0, is 44% and remain in the moderate risk for having CAC, while 56% are incorrectly classified as high risk. Abbreviation: CAC, coronary artery calcium.

strongly recommend patients with high risk and reduce the substantial number of unnecessary CAC score screenings (Fig 4, S2 and S4 Tables in S1 File). Like any radiographic investigations, the CAC score has drawbacks, including radiation exposure and incremental uncovered costs. Despite these limitations, CAC-prob, with its excellent discriminative ability (ORC 0.81), offers a reliable and cost-effective tool to support physicians' decisions.

## Strengths and limitations

This study represents a pioneering step in developing a CAC score prediction model (CAC-prob) for guiding CAC score screening recommendations. The model was developed based on ordinal logistic regression, featuring three outcome categories to offer a spectrum of predictive information. Additionally, our study included a balanced proportion of CAC score in each category.

There are limitations to be addressed. Firstly, the model was developed using retrospective data, which undeniably contains certain biases and missing data. Some of important predictors, such as family history of CAD, BMI were not included in candidate due to a poor documentation and unreliability. For the missing valued of included predictors, we employed MICE to impute the missing values and ensure that both statistical efficiency and uncertainty were maintained. Secondly, we included patients who may have already been prescribed a lipid-lowering drug or have a high baseline risk. This inclusion may limit the generalizability to patients who are statin-naïve or have a lower baseline risk. Thirdly, CAC-prob relies on static predictions, primarily hinging on a single modifiable predictor, low HDL-C. The gradual increase in age alone eventually guides patients to the CAC score recommendation cut-off point. However, with increasing age and consequently increasing ASCVD risk, the benefits of

CAC score screening become more prominent as it offers a precise prognosis and helps with preventive strategies. Lastly, it is important to note that our model was developed using data from a single center. To ensure the robustness of the model's performance, further domain-specific or multicenter externally validated studies should be conducted to address this limitation before implementing the model application in a broader clinical setting.

## Conclusion

This study developed the prediction model for recommending CAC score screening, CAC-prob, based on four routine clinical predictors: age, gender, the presence of hypertension or DM, and low HDL-C. The model has the ability to classify patients into three risk groups with excellent discriminative ability (ORC of 0.81). The adoption of CAC-prob has the potential to support clinicians in making decisions, facilitating patient access to CAC screening with the prospect of enhancing cost-effectiveness.

## Supporting information

**S1 File. Supplementary S1-S5 Tables.**
(DOCX)

**S2 File. CAC prob calculator.**
(XLSX)

## Acknowledgments

This study was partially supported by Chiang Mai University and Faculty of Medicine, Chiang Mai University.

**Declaration of AI and AI-assisted technologies in the writing process:** During the preparation of this work the author(s) used ChatGPT 3.5 in order to check and correct grammatical errors during the manuscript writing process. After using this tool/service, the author(s) reviewed and edited the content as needed and take(s) full responsibility for the content of the publication.

## Author Contributions

**Conceptualization:** Pakpoom Wongyikul, Apichat Tantraworasin, Pannipa Suwannasom, Tanop Srisuwan, Yutthaphan Wannasopha, Phichayut Phinyo.

**Data curation:** Pakpoom Wongyikul, Pannipa Suwannasom, Tanop Srisuwan, Yutthaphan Wannasopha, Phichayut Phinyo.

**Formal analysis:** Pakpoom Wongyikul, Phichayut Phinyo.

**Funding acquisition:** Phichayut Phinyo.

**Investigation:** Pakpoom Wongyikul, Apichat Tantraworasin, Pannipa Suwannasom, Tanop Srisuwan, Yutthaphan Wannasopha, Phichayut Phinyo.

**Methodology:** Pakpoom Wongyikul, Apichat Tantraworasin, Pannipa Suwannasom, Tanop Srisuwan, Yutthaphan Wannasopha, Phichayut Phinyo.

**Project administration:** Phichayut Phinyo.

**Resources:** Pakpoom Wongyikul, Apichat Tantraworasin, Pannipa Suwannasom, Tanop Srisuwan, Yutthaphan Wannasopha, Phichayut Phinyo.

**Software:** Pakpoom Wongyikul, Phichayut Phinyo.

**Supervision:** Phichayut Phinyo.

**Validation:** Apichat Tantraworasin, Pannipa Suwannasom, Tanop Srisuwan, Yutthaphan Wannasopha, Phichayut Phinyo.

**Writing – original draft:** Pakpoom Wongyikul.

**Writing – review & editing:** Pakpoom Wongyikul, Apichat Tantraworasin, Pannipa Suwannasom, Tanop Srisuwan, Yutthaphan Wannasopha, Phichayut Phinyo.

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
