## [Decision Letter · Decision Letter 0]

25 Jun 2024

PONE-D-24-20384Prediction Model for Recommending Coronary Artery Calcium Score Screening (CAC-prob) in Cardiology Outpatient Units: A development studyPLOS ONE

Dear Dr. Phinyo,

Thank you for submitting your manuscript to PLOS ONE. After careful consideration, we feel that it has merit but does not fully meet PLOS ONE’s publication criteria as it currently stands. Therefore, we invite you to submit a revised version of the manuscript that addresses the points raised during the review process.

**Please address all the comments made by peer reviewers before the manuscript would be reconsidered.**

We look forward to receiving your revised manuscript.

Kind regards,

Tom Wang

Academic Editor

PLOS ONE

Journal Requirements:

This study was partially supported by Chiang Mai university and faculty of medicine, Chiang Mai university.

The authors have declared that no competing interests exist

3. We notice that your supplementary tables are included in the manuscript file. Please remove them and upload them with the file type 'Supporting Information'. Please ensure that each Supporting Information file has a legend listed in the manuscript after the references list.

Additional Editor Comments:

Please adequately address all the reviewer comments for us to reconsider the manuscript.

Reviewers' comments:

Reviewer's Responses to Questions

**Comments to the Author**

1. Is the manuscript technically sound, and do the data support the conclusions?

Reviewer #1: Yes

Reviewer #2: Partly

2. Has the statistical analysis been performed appropriately and rigorously? 

Reviewer #1: Yes

Reviewer #2: I Don't Know

3. Have the authors made all data underlying the findings in their manuscript fully available?

Reviewer #1: Yes

Reviewer #2: Yes

4. Is the manuscript presented in an intelligible fashion and written in standard English?

Reviewer #1: Yes

Reviewer #2: Yes

5. Review Comments to the Author

**Reviewer #1:** Having a prediction model on whether a CAC score is warranted for patients is certainly appealing given the various guidelines out there. In this retrospective cohort, there seems to be good correlation using the model and predicting the CAC outcome. However, there are several limitations of the study.

First, this is a single center cohort and thus may not be applicable outside of a Thai population.

Second, as the authors acknowledge, including diabetic patients increases the ASCVD risk at baseline of the group. Diabetic patients are already known to have higher ASCVD risk and the condition itself is an indication to start statin therapy without the need of CAC scoring. Would it be better to remove diabetes out of the model and include just hypertension?

Lastly, please include how many patients are on statins. This is likely to affect your CAC scores with higher weighted CAC scores but lower ASCVD overall and thus may decrease the prognostic value of CAC.

**Reviewer #2:** Wongyikul et al. manuscript “Prediction Model for Recommending Coronary Artery Calcium Score Screening (CACprob) in Cardiology Outpatient Units: A development study” reported a retrospective cohort study who had CAC score performed, to come up with a risk model to predict higher CAC and :

1. Explain what is the exact outcome you have analysed to develop your model. Is it to predict patients at high risk of CAC>100 (or both moderate risk and high risk group)? Also, how exactly are patients classified into these risk groups, as by your methods it seems like it is not solely based on the actual CAC score reported.

2. CAC is a surrogate measure but not a clinical event – why not try to predict CV events/MACE, but rather predict CAC group?

3. How were the 8 candidate predictors selected? Why was diabetes/hypertension lumped as 1 group? What is angina chest pain included – CAC should be performed in asymptomatic patients? Why does lipid tests take up 3 factors? What about other risk factors like BMI/metabolic syndrome, smoking, family history of coronary heart disease?

4. Compare the performance your risk model (AUCs) with other established score such as PCE (ACC/AHA), FRS, SCORE, QRISK3 etc. at predicting the CAC outcome in your study. Need to demonstrate the AUC of your model is significantly higher (not just numerically, use statistical tests)

5. Risk scores typically performs best from the cohort it was derived from. Please apply the score to a more recent cohort from your hospital and assess performance for validation purposes.

6. How does your study results impact management? Who should have this risk model calculated? Should patients predicted at high risk get CAC, or be empirically treated with statins, or both?

7. Add to the limitations section what can’t be addressed above.

6. PLOS authors have the option to publish the peer review history of their article (what does this mean?). If published, this will include your full peer review and any attached files.

Reviewer #1: No

Reviewer #2: No

---

## [Author Response · Author response to Decision Letter 0]

29 Jul 2024

Reviewers comment on the manuscript 

Entitles: “Prediction Model for Recommending Coronary Artery Calcium Score Screening (CAC-prob) in Cardiology Outpatient Units: A development study” 

Dear Editor and reviewers,

 We would like to thank you for your valuable reviews and comments. It is our great pleasure to have an opportunity to revise our manuscript. We have revised and modified our manuscript with some additional information as suggested by reviewers’ comment. We hope that our revisions will improve the quality of the manuscript and give a clearer vision of research methodology to meet qualification for publication in PLOS ONE. Please inform us if further information or clarification needs to be addressed. We are looking forward to your reviews and would be grateful to make our response.

Reviewer #1: Having a prediction model on whether a CAC score is warranted for patients is certainly appealing given the various guidelines out there. In this retrospective cohort, there seems to be good correlation using the model and predicting the CAC outcome. However, there are several limitations of the study.

1. First, this is a single center cohort and thus may not be applicable outside of a Thai population.

Answer: We appreciate your feedback. We acknowledged this limitation and state in the limitation section in line 384-388 As follow “Lastly, it is important to note that our model was developed using data from a single center. To ensure the robustness of the model's performance, further domain-specific or multicenter externally validated studies should be conducted to address this limitation before implementing the model application in a broader clinical setting.”

2. Second, as the authors acknowledge, including diabetic patients increases the ASCVD risk at baseline of the group. Diabetic patients are already known to have higher ASCVD risk and the condition itself is an indication to start statin therapy without the need of CAC scoring. Would it be better to remove diabetes out of the model and include just hypertension? 

Answer: We appreciate your feedback and would like to address your concern. we agree that diabetic patient mostly start statin therapy due to high-risk baseline. However, our study aim is to develop a prediction model to enhance the appropriate use of CAC, did not directly guide the statin use. The CAC score can enhance the shared decision-making process through more accurate risk prediction. A significant advantage of the CAC score is its ability to reclassify risk both downward and upward [1]. In real-world practice, a doctor might be particularly worried about the statin therapy being too low for some patients, while for others, the concern might be avoiding overly stringent statin therapy. For instance, if diabetic patients have CAC scores ≥100, their risk can be reclassified, potentially leading to a more aggressive statin therapy approach and subsequent risk reduction [2-3]. Furthermore, the visual representation of individual atherosclerotic plaque burden provided by the CAC score can improve patient adherence to prescribed therapies [4]. 

We have stated the advantage of incorporating DM as predictor in line 344-351 as follow: “For our development set, patients with DM, accounting for 15% of the total population, were included in the analysis. Due to the inherently high atherosclerotic cardiovascular disease (ASCVD) risk among DM patients, most of them would probably be treated with statin therapy irrespective of their CAC scores. However, the added value of CAC score screening, as recommended by CAC-prob, lies in its ability to enhance discriminative ability and reclassify risk upward regardless of DM status [39, 40]. In cases where DM patients have CAC scores ≥100, there is an opportunity for risk reclassification, potentially leading to a more aggressive statin therapy approach and subsequent risk reduction [39, 41, 42].” 

Additionally, the model's discriminative performance was similar with or without including diabetes mellitus (DM) (Table X1). However, there are advantages to incorporating DM as a parameter. First, despite DM's low predictive ability in our cohort, combining DM with hypertension makes the model more effective and parsimonious (increase statistical power and reduce number of predictor in model) [5]. Secondly, the common barriers and challenges to be considered include clinician’s attitude and recognition to the model [6]. They may subconsciously doubt the validity of a model that excludes well-known predictors like diabetes.

Table X1 Comparison of Discriminative Performance: Model with DM excluded vs. DM included 

Discriminative ability Average Value 0 vs >0 <100 vs > 100 1-99 vs 100

HTN only 

Generalized ROC 0.73 0.61

Average ROC 0.76 0.79 0.73 

Ordinal C-index 0.82 

DM or HTN

Generalized ROC 0.73 0.60

Average ROC 0.76 0.79 0.73 

Ordinal C-index 0.82 

Reference 

1. Golub, I. S., Termeie, O. G., Kristo, S., Schroeder, L. P., Lakshmanan, S., Shafter, A. M., Hussein, L., Verghese, D., Aldana-Bitar, J., Manubolu, V. S., & Budoff, M. J. (2023). Major Global Coronary Artery Calcium Guidelines. JACC. Cardiovascular imaging, 16(1), 98–117. https://doi.org/10.1016/j.jcmg.2022.06.018

2. Malik S, Zhao Y, Budoff M, et al. Coronary Artery Calcium Score for Long-term Risk Classification in Individuals With Type 2 Diabetes and Metabolic Syndrome From the Multi-Ethnic Study of Atherosclerosis. JAMA Cardiol. 2017;2(12):1332–1340. doi:10.1001/jamacardio.2017.4191

3. Authors/Task Force Members; ESC Committee for Practice Guidelines (CPG); ESC National Cardiac Societies. 2019 ESC/EAS guidelines for the management of dyslipidaemias: lipid modification to reduce cardiovascular risk". Atherosclerosis 2019;290:140-205. https://doi.org/10.1016/j.atherosclerosis.2019.08.014.

4. Muhlestein JB, Knowlton KU, Le VT, Lappe DL, May HT, Min DB, Johnson KM, Cripps ST, Schwab LH, Braun SB, Bair TL, Anderson JL. Coronary Artery Calcium Versus Pooled Cohort Equations Score for Primary Prevention Guidance: Randomized Feasibility Trial. JACC Cardiovasc Imaging. 2022 May;15(5):843-855. doi: 10.1016/j.jcmg.2021.11.006. Epub 2021 Dec 15. PMID: 34922872.

5. Steyerberg E (2009) Evaluation of performance. In: Steyerberg E (ed) Clinical prediction models. a practical approach to development, validation and updating. Springer, New York, pp 270–279

6. Cowley LE, Farewell DM, Maguire S, Kemp AM. Methodological standards for the development and evaluation of clinical prediction rules: a review of the literature. Diagn Progn Res. 2019 Aug 22;3:16. doi: 10.1186/s41512-019-0060-y. PMID: 31463368; PMCID: PMC6704664.

3. Lastly, please include how many patients are on statins. This is likely to affect your CAC scores with higher weighted CAC scores but lower ASCVD overall /and thus may decrease the prognostic value of CAC. 

Answer: Thank you for your suggestion. According to our data, 132 patients have used lipid lowering drug and all of those patients received statin therapy at the time of data collection. We have added the text to provide the information according to your suggestion in line 194-196 as follow: “Approximately 40% of all patients were taking antihypertensive (40.8%) and lipid-lowering medications (36.7%). All patients on lipid-lowering drugs were receiving statin therapy.”

Reviewer #2: Wongyikul et al. manuscript “Prediction Model for Recommending Coronary Artery Calcium Score Screening (CACprob) in Cardiology Outpatient Units: A development study” reported a retrospective cohort study who had CAC score performed, to come up with a risk model to predict higher CAC and :

1. Explain what is the exact outcome you have analysed to develop your model. Is it to predict patients at high risk of CAC>100 (or both moderate risk and high risk group)? Also, how exactly are patients classified into these risk groups, as by your methods it seems like it is not solely based on the actual CAC score reported.

Answer: We appreciate your feedback and would like to address your concern. Our outcomes were the subcategories of absolute CAC score including CAC score 0, CAC scores 1-99, CAC scores ≥100. Based on those outcomes, the model was developed using the partial proportional odds model (ordinal logistic regression) with a stepwise backward elimination. Probability of each individual for having CAC score 0, CAC scores 1-99, CAC scores ≥100. was then calculated. 

For more clarify, we have edited the text in model development section in line 139-143 as follow “The model was developed using the partial proportional odds model with a stepwise backward elimination approach. The outcomes used in the model were CAC score 0, CAC scores 1-99, CAC scores ≥100. Initially, all pre-selected predictors were included. Subsequently, variables that were not significant at the level of 0.05 were excluded. The probabilities of being each category were estimated.” 

For the interpretation, the estimated probabilities for individuals having CAC scores of 0, 1-99, and ≥100 were classified into three risk categories based on selected cut points: (1) Low risk of having a CAC score above 0, (2) Moderate risk of having a CAC score above 0 but unlikely to be ≥100, and (3) High risk of having a CAC score above 0 and likely to be ≥100. For more clarity, we have revised the text in the cut-point selection section in line 166-180 as follows. “Our aim was to determine the cut point values that can effectively recommend patients to either undergo CAC screening or not, ensuring a balance between the potential downsides of overusing CAC score screening and its benefits. 

For practicality, we transformed the CAC score prediction model into a user-friendly web application called 'CAC-prob.' Once clinical and laboratory parameters are entered, the application calculates the probability of CAC score >0 and CAC score ≥100. Patients were then classified into three distinct ordered categories as follows: (1) Low risk of having a CAC score >0, (2) Moderate risk of having a CAC score >0 but unlikely to be ≥100, and (3) High risk of having a CAC score >0 and likely to be ≥100. Patients in the first group would not be recommended for CAC score screening and would follow routine standard care. Those in the second group would receive a recommendation for CAC score screening, with physician discretion or an increase in statin potency if they do not meet the reduction target. Patients in the third group would be strongly advised to undergo CAC score screening, along with an increase in statin potency if they do not meet the reduction target.”

2. CAC is a surrogate measure but not a clinical event – why not try to predict CV events/MACE, but rather predict CAC group?

Answer: We appreciate your feedback and would like to address your concern. We agree that predicting hard events yields more meaningful results. However, many previous studies have already established associations between clinical predictors like hypertension, diabetes mellitus, cholesterol levels, and others, and have developed risk stratification tools such as the Pooled Cohort Equation (PCE) and Framingham Risk Score (FRS) [1-2]. These are not the issues we aimed to address in our study. Our focus was on developing a prediction model to recommend whether patients should undergo CAC score screening.

The benefits of CAC score in risk stratification have been demonstrated in numerous studies outside our country. However, evidence in Thailand is limited. Recent studies assessing the added value of the CAC score on Thai cardiovascular (CV) risk have shown a significant improvement in the Net Reclassification Index [3]. According to the American College of Cardiology/American Heart Association (ACC/AHA) practice guidelines [1], the CAC score is currently endorsed as a risk stratification tool for patients with intermediate CV risk. However, it is not yet recommended in Thai clinical practice guidelines [4]. The use of CAC in our context mainly depends on the discretion of the primary doctor and patient accessibility. Consequently, many low CV risk patients may undergo unnecessary CAC testing, leading to low-value aggressive treatment or investigations. We advocate for the use of the CAC score as a risk stratification tool but in a more appropriate manner that balances CV risk with the potential for unnecessary radiation exposure and aggressive treatment.

Reference 

1. Grundy SM, Stone NJ, Bailey AL, et al. 2018 AHA/ACC/AACVPR/AAPA/ABC/ACPM/ADA/AGS/ APhA/ASPC/NLA/PCNA Guideline on the Management of Blood Cholesterol: Executive Summary: A Report of the American College of Cardiology/American Heart Association Task Force on Clinical Practice Guidelines. J Am Coll Cardiol. 2019;73(24):3168–3209.

2. Anderson TJ, Gregoire J, Pearson GJ, et al. 2016 Canadian Cardiovascular SocietyGuidelines for the Management of Dyslipidemia for the Prevention of Cardiovascular Disease in the adult. Can J Cardiol 2016;32:1263-82

3. Tiansuwan N, Sasiprapha T, Jongjirasiri S,Unwanatham N, Thakkinstian A, Laothamatas Jand Limpijankit T (2023) Utility of coronaryartery calcium in refining 10-year ASCVD riskprediction using a Thai CV risk score.Front. Cardiovasc. Med. 10:1264640.

4. The Royal College of Physicians of Thailand., 2016. RCPT clinical practice guideline on pharmacologic therapy of dyslipidemia for atherosclerotic cardiovascular disease prevention. Accessed June 30, 2021; http://www.thaiheart.org/images/column_1487762586/2016%20RCPT%20Dyslipidemia%20Clinical%20Practice%20Guide line.pdf

3. How were the 8 candidate predictors selected? Why was diabetes/hypertension lumped as 1 group? What is angina chest pain included – CAC should be performed in asymptomatic patients? Why does lipid tests take up 3 factors? What about other risk factors like BMI/metabolic syndrome, smoking, family history of coronary heart disease? 

Answer: We appreciate your feedback and would like to address your concern. 

We grouped diabetes and hypertension because, in our data, the predictive performance of diabetes alone was low, as shown by its similar distribution across all subcategories (in manuscript Table 1). Despite the low predictive performance in our cohort, the association between diabetes and CAC score may be strong in other settings, as demonstrated in previous studies [1-2]. Combining this predictor with hypertension enhances the model's generalizability and makes it more effective and parsimonious by increasing statistical power and reducing the number of predictors [3]. According to the Table X1, the exclusion of diabetes from the model had a negligible impact on its overall predictive performance. Furthermore, the common barriers and challenges to be considered include clinician’s attitude and recognition to the model [4]. They may subconsciously doubt the validity of a model that excludes well-known predictors like diabetes. 

Table X1 Comparison of discriminative performance: Model with DM excluded vs. DM included 

Discriminative ability Average Value 0 vs >0 <100 vs > 100 1-99 vs 100

HTN only 

Generalized ROC 0.73 0.61

Average ROC 0.76 0.79 0.73 

Ordinal C-index 0.82 

DM or HTN

Generalized ROC 0.73 0.60

Average ROC 0.76 0.79 0.73 

Ordinal C-index 0.82 

The other predictors you mentioned are indeed important and may enhance the predictive performance of the model. However, information on family history of coronary heart disease, height, and weight was either missing or not routinely and accurately collected. For metabolic syndrome, its diagnosis requires any 3 out of 5 conditions: elevated waist circumference (not routinely collected), elevated triglycerides, reduced HDL-C, elevated blood pressure, and elevated fasting glucose. Including metabolic syndrome as a predictor could cause statistical collinearity with other pre-selected predictors, reducing the statistical significance of other common predictors [3]. 

We have added our limitation regarding to this issue in line 373-377 as follow: “There are limitations to be addressed. Firstly, the model was developed using retrospective data, which undeniably contains certain biases and missing data. Some of importance predictors, such as family history of CAD, BMI were not included in candidate due to a poor documentation and unreliabilit

---

## [Editor Report · Decision Letter 1]

1 Aug 2024

Prediction Model for Recommending Coronary Artery Calcium Score Screening (CAC-prob) in Cardiology Outpatient Units: A development study

PONE-D-24-20384R1

Dear Dr. Phinyo,

We’re pleased to inform you that your manuscript has been judged scientifically suitable for publication and will be formally accepted for publication once it meets all outstanding technical requirements.

Kind regards,

Tom Wang

Academic Editor

PLOS ONE

Additional Editor Comments (optional):

Thank you for adequately addressing the reviewer comments.

---

## [Editor Report · Acceptance letter]

8 Aug 2024

PONE-D-24-20384R1 

PLOS ONE

Dear Dr. Phinyo, 

I'm pleased to inform you that your manuscript has been deemed suitable for publication in PLOS ONE. Congratulations! Your manuscript is now being handed over to our production team.

Kind regards, 

on behalf of

Dr. Tom Kai Ming Wang 

Academic Editor

PLOS ONE